# "Power to" for High Street Sustainable Development: Emerging Efforts in Warsaw, Poland

Artur Jerzy Filip 

Department of Urban Planning and Management, Faculty of Architecture, Warsaw University of Technology, 00-659 Warsaw, Poland; artur.jerzy.filip@pw.edu.pl

**Abstract:** Global discussions on the future of high streets, especially today in times of epidemiological, political, and market turmoil, emphasize the importance of high streets as laboratories for urban walkability, resilience, and sustainability. The major condition, however, is a collaborative, cross-sectoral approach towards high street development. Such efforts have been recently undertaken in Warsaw, Poland, to develop a lively but organized shopping street almost from scratch—a few promising joint initiatives with this goal have been undertaken in Warsaw over the last two decades. Building upon a broad document review and in-depth interviews with sixteen pioneers (business consultants, public authority leaders, and planning experts) directly involved in the development of high streets in Warsaw, this study reconstructs and analyzes their efforts in urban collaboration through the lens of Urban Regime Theory. By discussing strengths and weaknesses of the regime structuring process, this paper points at critical difficulties in high street sustainable development (and consequently, also to overall urban walkability, resilience, and sustainability) which are the inertia of mutual perception by stakeholders, dependency on singular leaders and their personal motivation, the necessity to reinvent the very idea of a high street anew, lack of adequate legal tools for cross-sectoral collaboration, and the stiffening effect of previously set guidelines.

**Keywords:** collaboration; public space; resilience; retail; streetscape; sustainability; urban regime



## 1. Introduction: The High Street Sustainability Question

High (main, shopping, commercial) streets, defined broadly as linear, primarily pedestrian public spaces combined with a mix of public and commercial services located along (with both open space and premises remaining under various managers and owners) are commonly perceived as manifestations of well-prospering, high-quality, and human-friendly urban tissue. As such, high streets reflect many urban dilemmas in an extreme way: commercial profitability vs. community values [1], uniqueness vs. convenience [2], locality vs. universalism [3], pedestrian-friendliness vs. good car accessibility [4], inclusiveness vs. safety control [5], or being a transition route vs. being a destination [6]. This all makes high streets a "particularly wicked problem" [6] (p. 18), as responsibility for high streets has been "disastrously fragmented"—with different stakeholders and agencies tending to their own narrow viewpoints and with barely anyone taking a holistic approach [6] (p. 18). This, however, makes the high street not only a problem in itself, but also a key part of the solution to broader urban problems—a kind of urban laboratory for walkability, resilience, and sustainability [3,6–11].

The "wicked" complexity of the high street sustainability problem is also multiscale. The global challenges are dependence on neoliberal markets and global scale corporations, transnational climate change policies, e-commerce developments, risk of public space commercialization, general lifestyle patterns shaped by pop culture and social media, and even recent restrictions on retail shopping due to the COVID-19 pandemic. Meanwhile, the locality-specific challenges are brick-and-mortar infrastructure, state and city legal frameworks, shopping habits, competition with nearby shopping malls, social and economic diversities, tourism, and historical heritage [1–6,12–18]. The core question stated by

recent research on high streets and their future has been on how to jointly envision and co-manage a long-term mission for each high street individually, regarding all the above, but focused mainly on urban sustainability [3,14], i.e., based on inherent social values, personal qualities of leaders, local history, and particular institutional culture. This in turn raises more specific questions about: (1) how to respond to existing urban constraints, (2) how to involve diverse high street stakeholders, (3) how to define a joint high street vision, (4) how to collaborate, and (5) and how to establish and achieve feasible goals.

Ideas about high streets and how to run them have evolved throughout history. Well-known solutions no longer work [14]. Common governance tools for managing high streets—Business Improvement Districts (BIDs) and Business Improvement Areas—were introduced over half a century ago in North America to provide basic streetscape improvements [17,19]. As the use of managerial policies [20] declined, BIDs came to be used as a panacea for urban regeneration [21–23]. Finally, BIDs spread globally as universal business management tools, duplicating many of the features of shopping malls [23] (p. 332), due to which the "BID Movement" fell into disgrace for imposing a logic that was criticized as too neoliberal and consumption-oriented [24] and leading to the undesirable commercialization of public space [23]. Today, scholars argue that BIDs as we know them are too narrow-oriented to meet the challenges of urban sustainability [25,26] and that new high-street strategies should be developed in a flexible, locality-specific way, with a particularly strong focus on local power dynamics and stakeholders' motives for involvement [25,27].

Accordingly, this study examines recent attempts to develop a lively but organized high street in Warsaw, Poland, where the initiatives had to be set up almost completely from scratch [13,16,28,29]. There has been a consensus among professionals, authorities, and residents that the downtown streets of Warsaw were in crisis and that the only way to revive them was to act strategically and cross-sectorally [13,15,16,18,30–32]. Over the past two decades, significant amounts of leadership energy have been put into high street development and various operating modes have been discussed, including BIDs [16,28,33], yet relatively few concrete results have been achieved. The hypothesis examined within this paper states that the pioneers directly involved in the process of high street development have significantly progressed their understanding of high streets as vehicles for urban walkability, resilience, and sustainability, even though the common desire and documents still call for an outdated vision. Today, COVID-19 restrictions and reports on generational changes among consumers have sparked renewed global and local discussions on high streets [31,34,35], yet as this paper finds, both government policies and public opinion have stuck to outdated modes of operation, while true pioneers must have dealt with particular difficulties to high street sustainable development.

## 2. "Power to" for High Streets

To understand where the true capacity for high street sustainable development lies, this paper examines the interdependent dynamics between structural urban conditions and multiple individual actions through the lens of Urban Regime Theory [36–41]. This perspective makes it possible to see high street development and operation as being socially produced and to seek action-oriented solutions based on collective effort, rather than expecting any authoritative "power over" [36] or individual "creative force" [42] to revive high streets on their own. The "power to"—conceptualized by Clarence N. Stone within Urban Regime Theory—stems from piecemeal interactions and the generation of new capabilities that lead to relatively sustainable collaborations over time. The concept of "power to" is therefore especially useful in framing "new things" for which no one has firm experience or conviction, and to which no one is hegemonic, yet the challenge is to craft arrangements that could develop along and advance those "new things" forward [36] (pp. 24–25), as it seems to be the case with today's high street and treating them as laboratories for urban sustainability.

Widespread use of Urban Regime Theory since the 1980s has made it a central concept in urban governance studies to this day, not only regarding U.S. cities but also enabling local and cross-national comparisons [41]. As the theory conceptualizes both formal and informal arrangements and encompasses both business and political endeavors [36,41], it is well-fitted to examine the developments of modern high streets. The concept has already been used to describe the reality of Polish urban development after the transition to a market economy in the 1990s [40] and to analyze the conditions of Polish shopping malls and high streets, in particular [43]. However, the flexibility and popularity of the theory come with the risk of "concept stretching" [37,41]. As not every form of urban collaboration or policy agenda makes for an urban regime, the concept must be used with a strong focus on its core criteria and must refer to (1) public–private (business included) coalitions, (2) sharing fragmented resources, (3) forming city-level policy agendas, (4) and achieving longstanding impact and change on the ground [37] (p. 829).

Even though Warsaw has not succeeded in forming an urban regime so far, the emerging efforts regarding high streets have touched greatly upon all four criteria mentioned above, and they could be examined as an emerging (neither shaped nor failed) regime [37], which has significantly affected the conditions for present and future high street development in Warsaw. Nowhere else but in this influence lies the progressive ambition and empowering potential of "power to" regime-building effort [36,39,41]. Urban Regime Theory does not posit that anyone has a clear or comprehensive vision, nor that any elites have free rein to pursue their goals [36]; what is posited is that the process of urban regime formulation sifts out glittery ideas and follows those with greater ease of execution [38], and that each subsequent minor opportunity agreed upon and exploited by the pioneers contributes to future long-term progression—or strengthens the status quo when the preference formation is too endogenous to existing power relationships [38]. This paper examines two decades of such steps taken by governmental and non-governmental pioneers adjusting (or polarizing) their preferences and understandings of what high streets in Warsaw should be and how they could be developed and contribute to urban sustainability.

## 3. Methods

The goals of this paper are (1) to reconstruct the sequence of recent efforts towards high street development in Warsaw, Poland, and (2) to find key difficulties due to which these efforts were not successful. Such locality-specific findings contribute to general academic discussion on (1) responding to existing urban constraints, (2) involving diverse high street stakeholders, (3) defining joint high street visions, (4) developing cross-sectoral collaboration, and (5) establishing and achieving feasible goals. At the same time, the findings allow for drawing very direct locality-specific conclusions for high street development in Warsaw, Poland.

For this purpose, a broad review of textual sources (municipal acts and documents, expert reports, research papers, reports from public discussions, and media coverage) was juxtaposed with personal testimonies of public authority leaders, business consultants, and planning experts who have been directly involved in high street development initiatives in Warsaw over the last two decades. The desk research and the interviews were carried out in parallel, as both parts of the research informed each other by pointing at new threads, issues, and persons.

The documents and the testimonies (referred to with a # in the text) were ultimately subjected to a narrative analysis (qualitative and hermeneutic) in order to reconstruct and examine the changing conditions, motives, and interaction dynamics of the "Urban Collaboration Efforts in Warsaw". These formal and informal attempts were then discussed as an exemplar of the emerging (neither shaped nor failed) regime, that has strongly affected the conditions for present and future high street development in Warsaw. By discussing locality-specific and interdependent challenges, this paper contributes to current debates and actions dedicated to high streets by pointing at critical "Difficulties to High Street Sustainable Development", and consequently to overall urban resilience and sustainability.

In the conclusion, this paper draws a lesson for possible high street developments in Warsaw by outlining missed but practical and feasible objectives to act upon.

The conceptual framework for analysis is based on Urban Regime Theory, which does not necessarily distinguish between leaders and their followers but aims to understand how diverse groups of pioneers influence one another and move forward across the five dimensions of the "power to" structuring process, according to which any collective urban action is undertaken within certain structural constraints (context), by diverse and unequal contributors (pioneers), adjusting their preferences and resources (middle ground) through a step-by-step feedback process (actions), up to the point where practical and feasible objectives can be realized (goals) [36] (see Figure 1).

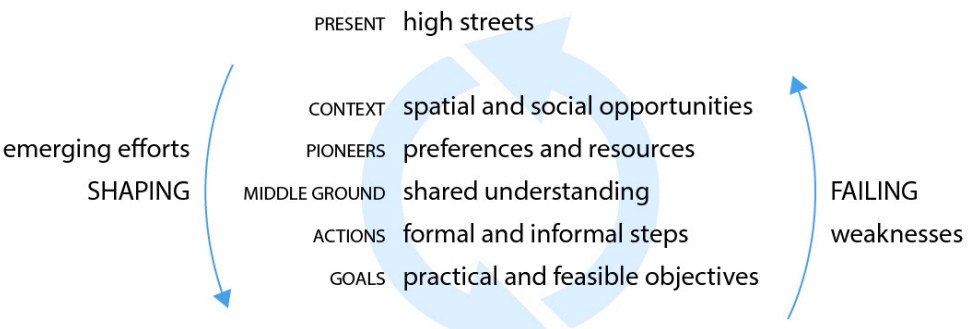

**Figure 1.** Conceptual framework for the Emerging (neither shaped nor failed) Regime for High Street Revival.

*Interviewing Protocol*

Candidates for interviewees—here termed "pioneers"—were selected based on their personal involvement and their leading role in initiatives aiming at high street development in Warsaw throughout the last two decades. Their professional background or formal status was not a determinant for the selection process; only afterward, the participants were divided into three groups: business consultants, public authority leaders, and planning experts. The first few interviewees were selected based on initial desk research, while the rest were selected using the snowball method. The selection was considered complete when no new person was identified as necessary for examination. Since most of the recent activities in Warsaw were more or less intertwined, the interviewees had some knowledge about other endeavors and their leaders (yet no one had a full understanding of high street issues in Warsaw). The whole 16-person group consisted of seven business consultants, five public authority leaders, and four planning experts; with a 9:7 male-to-female ratio (Table 1).

The interviews lasted from 50 min up to 2 h and were conducted in different places to ensure the interviewees' convenience and comfort. All the interviews were recorded and transcribed later on, and both the recordings and transcripts remained in the author's archive. The interviewees were informed about the goal of this research, the subjects to be brought up during the interview, the methods of data management and security, their personal anonymity, and the ways of using the findings in an academic article. All the rules met ethical standards for academic research involving human participants and were presented on paper and signed by each interviewee before the interview started.

All 16 semi-structured qualitative expert interviews were conducted in a flexible conversational mode and with openness to new threads brought up by the interviewees. Basically, each interviewee was asked five open-ended questions about the efforts: (1) What spatial and social opportunities determined the efforts? (2) Who was involved and what preferences and resources did they bring in? (3) What middle ground (if any) was agreed upon? (4) What formal and informal steps were undertaken? (5) What practical and feasible objectives were established? Additionally, each interviewee commented on the weaknesses

of each abovementioned element: (6) What and who was missing but would have been desirable for the efforts to be successful? Even though Urban Regime Theory served as a basis for the research conceptual framework, both in the way the questions were formulated and in the way the research analysis was conducted, the theory itself was not a subject of the interviews, and so no knowledge of Urban Regime Theory was expected or required of the interviewees.

**Table 1.** Summary of the interviewees.

| Id (#) | Date | Sector | Male/Female |
|---|---|---|---|
| 1 | 13 October 2022 | planning expert | F |
| 2 | 25 October 2022 | business consultant | M |
| 3 | 25 October 2022 | business consultant | F |
| 4 | 25 October 2022 | planning expert | M |
| 5 | 26 October 2022 | business consultant | M |
| 6 | 27 October 2022 | public authority leader | M |
| 7 | 1 November 2022 | business consultant | M |
| 8 | 7 November 2022 | business consultant | M |
| 9 | 10 November 2022 | planning expert | M |
| 10 | 16 November 2022 | business consultant | F |
| 11 | 16 November 2022 | public authority leader | M |
| 12 | 16 November 2022 | public authority leader | F |
| 13 | 23 November 2022 | public authority leader | F |
| 14 | 23 November 2022 | public authority leader | F |
| 15 | 6 December 2022 | planning expert | F |
| 16 | 8 December 2022 | business consultant | M |

The interviews were conducted in parallel with the desk research, as both parts of this research verified and informed each other by pointing at new threads, issues, and persons. Therefore, there is little risk of omitting and not interviewing any key pioneer, and the general narrative constructed in this paper can be considered certain. However, it must be admitted that any additional interviewee would have contributed to a better understanding of important details.

## 4. Results: Urban Collaboration Efforts in Warsaw

### 4.1. Local Context for High Street Development

*There is no true high street in Warsaw!*—This is one of three statements that has long been repeated by literally everyone involved: planning experts (interviews #1,4,9,15), public authority leaders (interviews #6,11–14), business consultants (interviews #2,3,5,7,8,10,16), and residents and local journalists [16,28,29], as well as by tourists and investors from abroad (#10) [29]. And whether the statement refers to the general crisis of Warsaw's downtown streets (#1,7), the characteristics of the urban tissue of post-war Warsaw (#11,12), or the absence of a high street tradition in the Polish capital (#9,11–13), with no local exemplary high street, the image of the desired one has drawn mainly on famous American and European examples [16,28,29]. Prestigious location, prominence, and the presence of global fashion brands have been commonly taken for granted as the key characteristics by which to evaluate the pace of high street development in Poland [13,16,29]. With time, the individual understanding of what constitutes a true high street has evolved (#1–3,8,12,15) but has not translated into a broad public debate in Poland on high streets as potential vehicles for environmental, social, or institutional resilience, not to mention any full-scale experiments on new modes of urban governance.

*High streets can only be revived by collaborative efforts!*—This is the second statement that has been repeated by everyone. The reasons for this are fourfold. First, a complex public–private ownership pattern (state government, district governments, housing associations, global holdings, and private owners, with some parcels still under restitution claims hindering long-term leasing agreements) requires all the stakeholders to shape tenant mixes collectively (#2,9,13–15) [13,15,16,44]. Second, the insufficient size and low physical quality of most downtown commercial properties (both interiors and storefronts being difficult to rebuild because of strong historical preservation rules) require everyone to agree on comprehensive architectural designs (#6–8,10,13) [16,30,43]. Third, the inconsistency and poor quality of the downtown streetscape (which started changing only recently after Poland's accession to the European Union) require a holistic approach to covering various issues: land use, blue and green infrastructure, car and pedestrian mobility, aesthetics, and visual communication (#3,6,10,13,15,16) [43]. And fourth, the absence of exemplary high streets in Warsaw and the need to build them practically from scratch require participants to come up with a joint vision that would be exciting for the leaders, profitable for the stakeholders, and intelligible to visitors (#1,10,14) [31].

*Warsaw lacks favorable conditions for cross-sectoral collaboration!*—This is the last of the three fundamental statements expressed by everyone, one regarding the location-specific conditions of Warsaw. First, for the past three decades, investment activities have been dominated by an entrepreneurial model of urban development, with a high level of competitiveness, selfishness, and focus on short-term profitability (#1–3), in which "corporate investors steamroll the obstacles" (#3) and benefit at the expense of the public interest (#11–13,15) [43,45]. Second, legal frameworks for urban planning are drawn-out and fragmented (#1,4,6,9), poorly regulate market-driven activities (#3,4,6,13), and fail to encourage strategic visioning (#1,4,7,13,16) [30]. Third, high levels of mutual prejudice and distrust, with contentious state politics in the background (#14), have led to a fear of public–private cooperation (#4,6) [46] and kept the private and the public "two separate realms" (#6,15,16). Fourth, with little tradition of community-based development (#1,16) and an absence of legal measures tailored to the particularities of institutional and business stakeholders (#3–5,11,13) [47], the participatory practice has been "all just a lot of hot air" (#4). And finally, public authority over downtown streets has been extremely fragmented, with over twenty city or district departments having an important say, and no single deputy mayor wielding the authority in one hand (#7,14), which has resulted in a lack of flexibility and decisiveness on the part of local government [35].

In the absence of prospering high streets, and with such unfavorable conditions for their development, Warsaw's retail market was easily satisfied with numerous modern shopping malls built in the 2000s, with Złote Tarasy in the very center of the city (see Figure 2). They have become the main showcase of modern Warsaw, where consumers have since spent most of their spare time and money [13,16,31,43,48], but permissions to open them downtown turned out to be an irreversible planning mistake (#1,2,4,6,8,12). The malls quickly introduced high-quality leasing and management standards, with legal and technical support, rent waivers, hard and soft fit-out, and long-term leasing agreements, not to mention constant advertising service and event programming (#2,3,10). The tenants have become "spoiled by the malls" (#10), while on the streets, "there was no one to speak to" (#2). Streets have "died" (#13), become "chaotic and disgusting" (#14), and are full of vacancies and contrasts (#6,7). What remained on the streets was, at first, banks and pharmacies, followed by a growing share of food and beverage [12,15,16]. With very few fashion brands and the decline of the few downtown department stores (with the very oldest one in Poland—Dom Braci Jabłkowskich—amongst others), the streets have come to be considered subordinate to the malls (#2,3) [15,43]. This trend has only recently begun to reverse in favor of local neighborhood-scale development (#2,3,10) [44,46,49]; nevertheless, shopping malls' capacity for reconfiguration and adaptation continues to make them an invincible rival in the shopping market (#2,9).

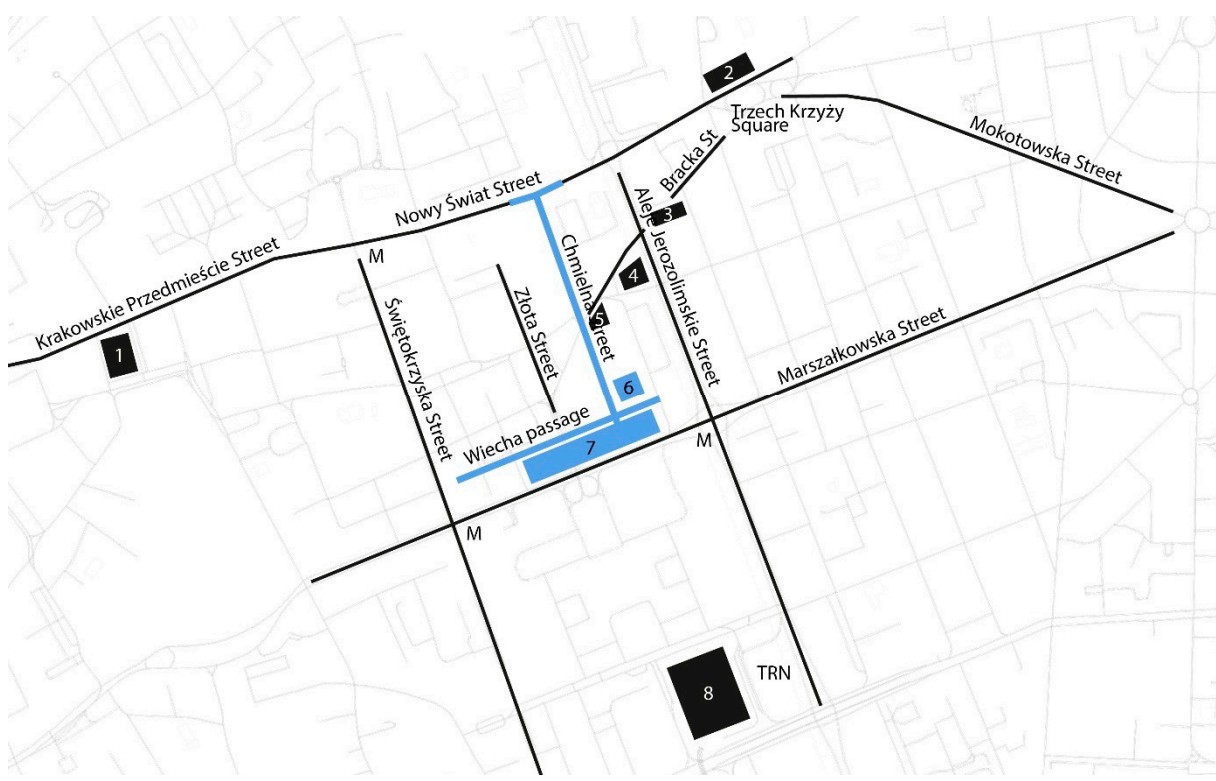

**Figure 2.** Streets and facilities downtown Warsaw, Poland, mentioned in the text: 1. Raffles Europejski, 2. Ethos, 3. Vitkac, 4. CEDET, 5. Dom Braci Jabłkowskich, 6. ZODIAK Warsaw Pavilion of Architecture, 7. Wars Sawa Junior, and 8. Złote Tarasy shopping mall; M—subway stations, TRN—Central Railway Station.

### 4.2. Two Decades of Collaborative Efforts

The closest to the high street ideal was Nowy Świat Street, where exclusive brands first appeared after the political and economic transformation of 1989 (#1,5–7,13), and where over a hundred "active entrepreneurs" formed an association with the goal of buying out the occupied properties (#5). They were the first high-street initiative in Warsaw (#7) and became a model for others (#5). Even after their re-privatization plan had failed (#5), they turned the initiative into a "friends of the street" group lobbying to facilitate administrative procedures against administrative red tape and to lower rents. The group also took over the organization of public events, including the street's iconic annual Christmas decorations (#5,7,10), and lobbied for limits on regular car traffic, making Nowy Świat a popular pedestrian promenade (#5). Around 2010, the street hosted such brands as Dior, H&M, and Orsay, along with the iconic bookstore Empik, and Krytyka Polityczna, an intellectual and social gathering place. However, it failed to maintain its presence for more than a decade (#5,10,12) [50] and has since turned into a mid-range dining street (#5,10) [28]. No "critical mass of shopping" was ever achieved (#11); people were more interested in window shopping (#10), and the stores suffered "dramatic losses" (#2,10,12). The burned-out leaders admit they failed to make Nowy Świat a prestigious high street (#5,10): the uncoordinated practice of renting short-term to the highest bidder (#5,10) led to speculative practices that further exacerbated the issue of random occupants and vacancies (#10).

Around 2010, when Nowy Świat was still in its prime, the city turned its attention to Trzech Krzyży Square (#1,3,11–13), which—compared with other stretches of the historical Royal Route—had strong shopping potential (#1,3,7) [51]. But after plans spearheaded by municipal authorities were narrowed down to the "aestheticization of the square" (#3), local stakeholders (owners and tenants, business, public, and not-for-profit organizations supported by independent experts) formed a grassroots coalition and developed their

own vision for the transformation of the square (#1,3,13). Even though their "motivations differed significantly" (#3), they succeeded in finding a middle ground defined primarily by the "diversity and abundance of people" in the square (#3) [51]. The coalition worked as an informal group bound by good relations between individuals, with no financing, only some barter exchange, and with no management structure: the vision was "meditated jointly by everyone" (#3). Their intention was just to spark public discussion and make the city more dedicated to the square's revitalization (#1,3) [51]. The vision was finally published in the form of a brochure and submitted, together with a list of detailed comments, as part of a municipal zoning procedure, which the coalition decided to rely upon in the absence of any other dialogue platform (#1,3). The partners also discussed the possibility of formalizing their collaboration, co-managing the tenant mix, and sharing sales results and even floated the idea of redistributing profits in the future. However, these were "casual conversations in casual settings, and were never carried out" (#3). The efforts turned out to be fruitless; the coalition has not found understanding among city officials, nor has the city accomplished its development plans to this day (#1,3,12,13).

In a situation where city officials have had no legal basis to enter into dialogue, negotiate contracts, or issue any profiled lease decisions (#4,8,10,12), and where private owners have rented without seeking institutional help (#10), no big brand was interested in opening on the streets (#2,10,15)—"it is a waste of time for them, too much effort, too much red tape" (#10). However, there have been four groups of tenants interested in the streets, even while the organizational, architectural, and streetscape standards were far from ideal. First, convenience stores and services (e.g., grocery stores, coffee shops, bakeries, barbers, etc.), which rely mainly on simple urban factors such as residence density and foot traffic, easily adapt to existing architectural conditions and are profitable in the short term (#3,5–8,10,11,15). Second, restaurants, even though they require specific technical facilities (ventilation, deliveries, waste storage, and disposal), thrive upon the streetscape by attracting passers-by and offering outdoor seating (#2,3,10,11). Third, small concept stores run by individuals who are prepared to devote personal time and resources to inconvenient procedures just for the opportunity to experiment with their niche ideas (#2,10). And fourth, luxury brands that "would never open in a shopping mall" (#10), so even if they are highly selective about their surroundings, they have no alternative to streets whatsoever, and must therefore find a luxurious oasis downtown or just give up on the market entirely (#10). These tenants make up the chaotic image of Warsaw's streets. Still, a surprisingly consistent exception is Mokotowska Street, which offers a neglected but classy architectural setting (#2,3), perfect for a "Saturday morning walk" (#2), and despite the lack of sufficient infrastructure and holistic management, boosted only by a few high-end brands that appeared in the nearby Trzech Krzyży Square (#2), Mokotowska became "a sure-fire hit" (#12), quickly attracting individual boutiques (#2) and spontaneously turning into Warsaw's top fashion street (#1–3,7,8,10–13) [31].

Streets widely considered to be the most promising for shopping—Marszałkowska, Chmielna, Świętokrzyska, and Jerozolimskie Avenue—still have not reached the volume, density, or quality required to be labeled high streets (#1–3,6–8,10–12,16) [13,16]. A few property management companies have attempted to collect at least a couple of properties and manage them jointly, but these efforts have been limited in scale (#2,6,8,10). However, there have been also few large single-owner commercial buildings whose retail properties were accessible from sidewalks and thus formed short stretches of prosperous high streets (#1–8). In the case of Raffles Europejski and Ethos, the owners offered classy urban surroundings and "clear visions and real commercialization processes" (#7) tailored to premium brands, which would not otherwise open locations, as they attach great importance not only to market indicators but also to the setting and "being right next to each other" (#10). In the case of Wars Sawa Junior, even though it loses out to regular shopping malls due to the lack of parking (#7,10), the location on Marszałkowska Street, right between two subway stations, guaranteeing 60 million passers-by a year (#2,14) and great visibility for millions of downtown drivers (#10), enticed top fashion chain brands to open their flagship

stores there, even if partly for promotional and advertising goals (#10). In both cases, while "shopping-mall-style" management made it possible to make the deals (#2,8,10), it was the street context itself—either classy or mass-market—that was the key advantage for the tenants (#2,7,8,10). At the same time, managers of close-by retail/business buildings on Bracka Street (newly renovated CEDET and newly built Vitkac) have operated with no attachment to public space whatsoever, the latter even demolishing the historical continuity of public space [52].

Since the city of Warsaw owns and operates not only public spaces, including streets, but also over three thousand downtown commercial properties (up to 30–40 percent on certain streets) (#6,7,12,14) [53], the local government was expected to play a significant role in shaping high streets in Warsaw (#1,6–8,10,14,16) [13,16]. Several professional and public debates on the issue have been conducted since 2013 as part of the land-use planning procedure, as a result of which some streets were designated as requiring particular spatial arrangements, with their ground floor premises designated for commercial use and their storefronts for reconstruction (#12,13) [54]. However, the plan has yet to be adopted, and even if it were adopted, any decision regarding the buildings would have to be approved by the conservation department (#13). Another way to regulate and manage downtown streets could be through a Local Revitalization Plan, which has never been attempted in Warsaw (#4). The general intention for high street development contributed to a new comprehensive plan of Warsaw (under preparation; #12,13) and to the New Warsaw Centre vision plan introduced by the mayor of Warsaw in 2019 with the clear intention—yet still no legal tool for effective implementation—to revive the high streets [55]. Indeed, the only governmental bodies with power over ground-floor properties are the myriad district departments tasked with managing municipal real estate (ZGNs), for whom commercial properties are just not a strategic issue (#6,7,12–14)—they "don't have any vision for it" (#13); in some cases, these departments actually oppose high street development, as their main concern is the well-being of local residents: "they will always stand on the side of peace and quiet" (#7).

The need for all stakeholders to undertake comprehensive action in a concerted manner has been discussed in less formal ways for years (#3,5,9), including by private developers seeking alternatives to shopping malls (#4,6,9,15,16) [33]. Concurrently, the challenge was taken up by the city, when two of its departments—the Economic Development Office and the Architecture and Spatial Planning Office—run by directors with cross-sectoral experience and backed up by the deputy mayor of Warsaw, launched a platform for cooperation between the city and business (#6,12,14), to create something "that operates in every city but Warsaw" (#14). In 2017, they initiated a series of cross-sectoral workshops for public and corporate sector representatives, with street retail being one of the subjects under discussion (#6–8,12,14,16) [28,53]. Even though both sides had previously had bad experiences (#6), the city's highest authorities met with the CEOs of a few dozen large corporations to discuss their common ground: "the desire to revive the city center" (#6,14). Both sides contributed presentations and discussed emerging issues (#6,12,14,16). Even though the city was perceived by business partners as talking too much (#16), not listening enough (#8), and being excessively cautious and hiding behind rules and regulations (#8), while, on the other hand, the business sector was perceived by city officials as too demanding, narrow-minded, and full of resentment (#11,12), the mood was very good and optimistic (#6,14,16): "Everyone was saying the same thing: why don't we have a high street in Warsaw?" (#14).

In a quest for solutions, the participants discussed three organizational matters. First, more collaboration was needed, so even if there is no legal basis for BIDs in Poland, some less formal partnership could be set up—a sort of "cross-sectoral task force" (#6,16). Second, more leadership was needed, so the city would need to appoint someone to manage the streets and move the issue forward—some kind of "high street manager" (#12). And third, more focus was needed, so a new "independent entity" would need to be established to manage high street properties according to jointly established policies and rules (#6,14).

The city promised to "think about it", but nothing much happened (#16). What the city in fact did was commission two recommendation studies. One, conducted by EY on the subject of managing commercial properties—but concerning only streets included in the revitalization program [56]—led city hall to introduce legal changes allowing for profiled bids, longer leasing agreements (up to 10 years), and subletting properties by so-called "rental operators" (up to 10 properties) (#6,8) [57]. And the second, conducted by JLL strictly on downtown high streets, identified some ready-to-use co-management tools, but, more importantly, examined the potential of a dozen streets and ultimately identified two locations—part of Marszałkowska Street and part of Jerozolimskie Avenue—for pilot projects (#6,8,14–16) [15].

The first, and so far last, chance to experiment—although in a manner contrary to JLL's street selection recommendation—came as early as 2018, when a global real estate management company purchased Wars Sawa Junior and expressed its intent to create a high street quality public space in collaboration with other stakeholders in the vicinity (#7), its readiness to devote financial resources (#2,4,6), and that it was already commissioning independent studies on the social context and feasibility of such an endeavor (#6,15). The city also was proactive (#2,3) and wanted to set up a coalition for downtown Warsaw; it was even willing to experiment with BID-like solutions (#12). This presented an unexpected opportunity, and both sides expressed their willingness to cooperate (#2,6,7,11), but they still lacked a mechanism to establish a legally binding arrangement (#11,12). The proposals were discussed, but the city was reluctant to make any informal commitments (#6,7), while the company treated public space improvements "with no missionary zeal" (#15), but as pure leverage for their commercial goals (#11,15). Thus, even though both sides shared a common ground, they could not agree upon the means and goals, and the discussions reached a dead end (#2,6,7,11,12). The problem of mutual misunderstanding constantly resurfaces, because while every attempt to collaborate brings both parties closer together, each failure reinforces the distrust between them: "the atmosphere for an honest dialogue has yet to be built" (#16).

JLL's study inspired a few more follow-up public and expert discussions [32,58] but was ultimately "shelved" (#8,14,16), partly due to the COVID-19 pandemic, and partly due to a lack of people, energy, and resources (#11,12,14). When the city organized architectural design competitions for the redevelopment of Chmielna and Złota Streets in 2021, attempts to collaborate with local stakeholders were already over, so the city provided the contestants with some high street recommendations [52] but put aside its former collaborative aspirations (#7,11). The announcement of the pilot project was copied and pasted into a working version of another city program on public spaces [47,59], thereby preventing deeper discussion, and instead simply repeating JLL's proposal, with its four serious weaknesses: unnecessarily restrictive criteria for prospective high streets—based on administrative names instead of on spatial–functional systems [52]; insufficient support in legal planning documents (#4); overly narrow identification of required stakeholders (#4); and excessive focus on fashion brands as constitutive elements of high streets (#15). There has been little excitement among city officials about that goal (#11,12): "it was sexy in the beginning, but then years passed, and it lost its appeal" (#6). The same was true of the private sector: in the early stage of high street development, commercial motivation is not enough: "there must be something more than profit motives—some madness, passion, love" (#1).

## 5. Discussion: Difficulties to High Street Sustainable Development

The juxtaposition of publicly accessible documents with personal testimonies of high street development pioneers proved how much the desires of the general public and the municipal guidelines are at odds with actual development dynamics and needs. The detailed findings are discussed and generalized below through the lens of Urban Regime Theory and across the five dimensions of the "power to" structuring process. There are five

critical difficulties to high street sustainable development and, consequently, also to overall urban walkability, resilience, and sustainability:

- Inertia of mutual perception by all the stakeholders (5.1.);
- Dependency on singular leaders and their personal motivation (5.2.);
- Necessity to reinvent the very idea of a high street anew (5.3.);
- Lack of adequate legal tools for cross-sectoral collaboration (BIDs does not play that role anymore) (5.4.);
- The stiffening effect of previously set guidelines (5.5.).

*5.1. Vicious Circle of Perception (Context)*

All the efforts undertaken in Warsaw over the past two decades have proved that both public and private sectors have (1) dealt with cross-sectoral coalitions (Nowy Świat Association, Trzech Krzyży Coalition, Wars Sawa Junior's attempts); (2) shared resources (organizing events, sharing know-how); (3) influenced city-level agendas (the round table, EY and JLL studies, and the recent programs for public space improvement); (4) and worked towards longstanding impact on the ground. Therefore, even though the "power-to" forming process has not been successful, the efforts shall be discussed as an "emerging regime" [37]. Neither shaped nor failed, the process has defined a tangled context for further efforts: The business sector has not opened high street stores, as there was no sufficient number of consumers, no professional management, nor any long-term vision on the part of the city. Consumers have not come to perceive downtown streets as desirable shopping locations, as the commercial offer and the quality of brick-and-mortar locations were too low. The local government has not undertaken major projects, as there was little certainty whether convenience-seeking customers and profit-oriented businesses (both already satisfied with shopping malls) would find them attractive. The efforts towards balanced, sustainable development did not fail but were hindered by the existing perception of the high streets, creating a vicious circle of passivity [38].

*5.2. Discouragement of Stakeholders (Pioneers)*

Throughout the two-decade-long process, the pioneers learned about their preferences and adjusted to some extent, mostly by limiting their expectations towards others, which was key for achieving and implementing a joint sustainable policy. However, some differences seemed insurmountable, and there was no stakeholder who could effectively assume a leadership role. The experience has shown that the business side was not the best party to reinitiate the dialogue since they were quickly accused of favoring particular interests, while the community side was too enigmatic (customers) at a large scale, and too NIMBY (residents) or too individualistic (landlords) at the local scale to play a regulatory role. Therefore, the impulse was expected to come from the government side, which could provide the legal basis and long-term public responsibility for the process. The city, on the other hand, remained fragmented and would need to expend significant effort and resources to restructure, which was a fundamental obstacle from the perspective of regime forming feasibility. So far, in the face of these difficulties, the pioneers once "obsessed with high streets" (#1)—no matter which sector they represented—felt powerless, became discouraged, and turned their attention to other issues, yet all of them hoped to become involved again if new opportunities or resources came their way (#1–16).

*5.3. Ambiguity about the Meaning (Middle Ground)*

As soon as the issue is back on the agenda, there will be probably little debate about learning from well-prospering shopping malls regarding high streets' spatial arrangement (edges and anchors), directing the footfall, improving streetscape standard and walkability, conceptualizing the tenant mix, supporting the tenants, programming common spaces, and advertising. Yet, there is still great ambiguity about the desired ends—the real meaning of high street sustainability. The corporate fashion brands have been already taken care of by the shopping malls and "won't be a driving force for Warsaw high streets" (#8).

Contrary to initial expectations, the experience has taught the pioneers that it was pointless to counteract the trends: if dining won over fashion, that was something to be accepted; the idea of high streets must have adapted. Consequently, due to the impossibility of developing a classic high street, the current "middle ground" has been evolving towards existing potentials: dining, startups, and popups, and has become much more open-ended, heterogenous, and egalitarian: "It will be something different in Warsaw. Some kind of a hybrid" (#3). This way, out of the "wickedness" of the high street issue, the Warsaw pioneers have started to perceive high streets more as laboratories for urban walkability, resilience, and sustainability—urban vehicles bringing together the challenges of economy, society, and governance—and finally more in line with the prevailing global trends in this area (#3,7,15,16) [3,6].

### 5.4. Hodgepodge of Experience and Knowledge (Actions)

No downtown street has turned into a democratic, egalitarian, and sustainable urban space so far, even though everyone claimed to support such a vision to varying extents. It turned out to be too "glittery" [39] to be implemented right away. Single-owner developments (Wars Sawa Junior, Raffles Europejski, Ethos) ended up becoming one-dimensional commercial spaces. Stakeholders who pursued public–private partnerships were spooked by the prospect of informal collaboration and fell victim to poorly designed regulations that paralyzed the city government and tethered it to the neoliberal logic of the market. The experience of non- or loosely coordinated actions proved that high streets self-developed only as an exception (Mokotowska street), in spite of pioneers' intentions (Nowy Świat, Chmielna streets), and mostly ended up as messy spaces with random tenant mixes and high vacancy levels. After a few collaborative initiatives, the regime-forming process stopped mid-way, and the general result has been little more than a hodgepodge of experience and knowledge, with no cross-sectoral forum where the preferences and ideas could be further negotiated, not to mention any legal vehicle to actually manage the process. Contrary to what Stone suggests, the model "power to" structuring process did not grow out of soft power itself, and it would need some hard power ("power over") to begin with in order to remove obstacles and to balance market forces, if not to ensure the sustainability of the further process.

### 5.5. Derivative Nature of the Guidelines (Goals)

A number of visions for high streets, and methods for defining their goals, were attempted collaboratively, discussed at a round table, and finally drafted by the JLL study into guidelines for a pilot project, the specifics of which were copied verbatim into subsequent documents. But after the collaborative efforts ceased, only a few city-driven investments into downtown streets continued, some of them only loosely related to the needs of future high streets (Chmielna Street), and others even clearly against them (Trzech Krzyży Square). Some of the improvements, such as sidewalks, greenery, benches, intersections, parking spaces, etc., were the "fundamentals" (#3,5,9) that would have needed to be completed anyways and could attract some stores to the streets, but on the other hand, it might turn out that these expensive changes would just have to be scrapped in 5–10 years (#2). In fact, the only reason these particular architectural reconstructions were carried out was because, at the time, they were the only ones that could feasibly be executed, even if they were performed unilaterally and with disregard for all other concerns [39]. In fact, some operational tasks, such as collecting big data on downtown streets (#7) or enforcing a zero-vacancy policy (#6) could be undertaken right away, if only to lay the groundwork for future activities, but these tasks were not specified in the guidelines. The seemingly handy guidelines for the pilot project, left undone yet formally reinforced, restricted further discussions and became almost a curse against high street sustainability instead of a blessing.

### 6. Conclusions: Missed but Feasible in Warsaw

Despite—and due to—the fact that every attempt so far has run into a dead end and most of the pioneering energy has burned out, this wealth of experience has prompted an evolution in the vision of the desirable high street towards an even more locality-specific and sustainable one. This paper substantiated the hypothesis that throughout the last two decades, the pioneers directly involved in the process of high street development in Warsaw have significantly progressed their understanding of high streets as vehicles for urban walkability, resilience, and sustainability, even though both government policies and public opinion had stuck to outdated modes of operation, which has remained the source of particular difficulties to high street sustainable development. Now, if the collaborative efforts for high street sustainable development are to regain momentum, the action plan should be jointly discussed and reformulated with an even stronger emphasis on what is feasible, down to earth, and with a critical but positive stance towards previous achievements—almost as if the revival process were well underway. A list of practical guidelines could include the following:

1. To stop equating a potential high street with any administratively designated street, and instead define a high street based on where its key elements (tenant mix and streetscape quality) are already located.
2. To reinitiate the collaborative effort by an entity that is already embedded in municipal power structures, has a broad business and public profile, and due to its physical location, is irrevocably bound to a target high street.
3. To appreciate the existing tenant mix and development dynamics, and to seek and amplify a hybrid high street rather than any narrowly defined vision.
4. To bring knowledge, experience, and people together within a public not-for-profit think-tank, ultimately with a legal basis for a decision-making process (anything from a multilateral contract up to a Local Revitalization Plan).
5. To take a few steps back and reopen the discussion anew.

While this may sound similar to a wish list, there is one place in Warsaw where all five guidelines could be implemented right away (see Figure 2, blue mark). Only instead of on a single street, the focus would have to be on an area of Wars Sawa Junior–Chmielna Street–Chmielna/Nowy Świat intersection (context). When examined in this way, the whole stretch turns out to consist of a broad mix of global fashion brands, food and beverage places, pop-up concept stores, cultural sites, tourist attractions, and dense residential fabric that are already forming a hybrid street (middle ground). One of its anchors is a municipal and mission-oriented creative hub, the ZODIAK Warsaw Pavilion of Architecture (pioneer), where not only city authorities and business leaders, but also independent experts, artists, activists, and residents could launch another set of debates not just on contemporary public and commercial developments but also, and to an equal extent, the historical identity of the shopping streets. Moreover, this would be nothing more than an ad hoc redefinition of existing guidelines (goals), just grounded more firmly in the streets' existing potential. And even though only through further collaborative effort will the soundness or weakness (feasibility) of these guidelines be confirmed, adapted, or redefined either way, future initiatives will contribute to the formation of a "power to" for Warsaw's high street sustainable development. This in turn will contribute to overall urban walkability, resilience, and sustainability.

**Funding:** The publication was funded by the Department of Urban Planning and Management, Faculty of Architecture, Warsaw University of Technology.

**Institutional Review Board Statement:** The study was approved by the Warsaw University of Technology Committee for the Ethics of Research Involving Human Participants (24 January 2024).

**Informed Consent Statement:** Informed consent was obtained from all subjects involved in the study.

**Data Availability Statement:** The interviews transcripts are not publicly available, as they contain information that could compromise the privacy of research participants.

**Conflicts of Interest:** The author declares no conflict of interest.

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
