# Peer review of "“Power to” for High Street Sustainable Development: Emerging Efforts in Warsaw, Poland"

_sustainability, doi:10.3390/su16041577_

Round 1
Reviewer 1 Report
Comments and Suggestions for Authors
The paper presents a comprehensive analysis of the challenges, efforts, and outcomes related to the sustainable development of high streets in Warsaw over a period of two decades. Here are some strengths and weaknesses of the paper:
S
1. The paper offers a detailed examination of various aspects influencing the sustainability of high streets, including public-private partnerships, stakeholder involvement, perception challenges, and evolving trends.
2. Rich Data and References: It draws upon a wide range of sources, including studies, reports, and expert opinions, to support its analysis and conclusions. This indicates thorough research and provides credibility to the findings.
3. Longitudinal Perspective: By spanning two decades, the paper provides a longitudinal perspective on the evolution of efforts towards high street sustainability in Warsaw. This long-term view allows for a deeper understanding of the challenges and trends.
4. Insightful Recommendations: The paper concludes with practical recommendations for revitalizing high streets, grounded in the analysis presented earlier. These recommendations offer valuable insights for policymakers, urban planners, and other stakeholders.
W
1. Limited Focus: While the paper offers a detailed examination of the situation in Warsaw, it may lack generalizability to other contexts. The findings and recommendations might not be directly applicable to high streets in different cities or regions with distinct socio-economic conditions.
2. Complexity of Analysis: The paper delves into various factors influencing high street sustainability, which can sometimes make the analysis complex and difficult to follow. Simplifying the presentation of findings could enhance clarity and accessibility for readers.
3. Limited Discussion of Stakeholder Perspectives: While the paper acknowledges the importance of stakeholder involvement, it may provide limited insight into the perspectives of different stakeholders, such as local businesses, residents, and government officials. A more in-depth exploration of these perspectives could enrich the analysis.
Overall, the paper offers a valuable contribution to the understanding of high street sustainability in Warsaw, although there are areas where further refinement and exploration could enhance its impact and applicability.
Author Response
Dear Reviewer,
Thank you for all your inquisitive remarks and comments. I did my best to take all of them into account and I believe that responding to your advice and requests made the paper better. All the changes have been documented with the “track changes facility” in Word.
Regarding your comments:
weakness #1 - I made the 5 main findings clearer and listed them in the discussion part and in the abstract; this way the universal findings are clearly distincted from the local-specific conclusions included in the conclusion part.
weakness #2 - I believe that claryfiying the five main findings adresses your comment,
weakness #3 - I must admit that a deeper discussion of the stakeholders' standpoints would had require another research paper, however that would be an exciting and interesting endeavor to be brought up,
Yours sincerely
Reviewer 2 Report
Comments and Suggestions for Authors
The Article titled ‘Power to” for High Street Sustainable Development: Emerging Efforts in Warsaw, Poland' is dealing with two main questions: first, the reconstruction of the sequence of recent efforts towards 135 high street development in Warsaw, Poland; and second, the identification of reasons behind failer of 136 efforts. This study analyses the efforts in urban collaboration through the lens of Urban Regime 17 Theory.
While the research topic is current and the manuscript follows the proposed structure of the journal, certain modifications are necessary to enhance scientific rigor, clarity, and overall quality of the document. Recommendations are provided to address these aspects.
The Abstract requires a rewrite to align with the journals guidelines (structure), it is meaningful to present the results of your own research.
The manuscripts’ relevance should be explained by citing appropriate new publications also within the last 5 years. Controversial and diverging hypothesis are missing or not well presented, and the research gap should be more clearly identified. Finally, the main aim and main conclusions of the own research should be highlighted.
In the Literature review, references should be upgraded with recent literature from the last 5 years.
Method section – are there any research hypotheses to be demonstrated and need to be tested (gap in research literature, knowledge). Not sufficient details are describe regarding the method (135-142):
“For this purpose, a broad review of textual sources (municipal acts and documents, expert reports, research papers, reports from public discussions, and media coverage) has been juxtaposed with personal testimonies of public authority leaders, business consultants, and planning experts who have been directly involved in high street development initiatives”
How can this part be replicate when necessary to justify the results of the manuscript? Describe the protocol more in detail. The use of the snowball principles is acceptable, but it need clarification, including why it is chosen and what are the disadvantages of this method.
Research results and figures are appropriate in a way to be easy understand. It is recommended to provide a more in-depth discussion. Defined hypothesis should be more explained how your results contribute to the existing body of knowledge in your field. In this case, your own hypotheses that are verified, and those that are not, must be included.
Author Response
Dear Reviewer,
Thank you for all your inquisitive remarks and comments. I did my best to take all of them into account and I believe that responding to your advice and requests made the paper better. All the changes have been documented with the “track changes facility” in Word.
Regarding your comments:
1. the abstract has been improved to include the findings,
2. the reference list has been improved, all the newly added references are from within last 3 years,
3. as to the research gap, the hypothesis, and research questions: they are explained and stated in the introduction part: "the core question" and research questions (lines 47-54), the the gap in research in practice (lines 55-68).
4. the "method" section has been substanitally improved and the "interview protocol" is now explained in much more detail.
5. the main 5 findings are now clearly listed and clearly distincted from the local-specific conclusions included in the conclusion part.
Yours sincerely
Round 2
Reviewer 2 Report
Comments and Suggestions for Authors
The criterion considered by the reviewer during evaluation is wheter the research design, questions, hypotheses and methods clearly stated. It is noted, that hypohesis are still missing.
Author Response
Dear Reviewer,
Once again thank you for your advice. Now I added clear expressions of the research hypothesis in the beginning (lines 98-101) and in the conclusion section (lines 631-636),
I hope now it will be more clear for the readers,
Yours sincerely